# Four-Way Decomposition of Effect of Alcohol Consumption and Body Mass Index on Lipid Profile

**DOI:** 10.3390/ijerph182413211

**Published:** 2021-12-15

**Authors:** Chaonan Gao, Wenhao Yu, Xiangjuan Zhao, Chunxia Li, Bingbing Fan, Jiali Lv, Mengke Wei, Li He, Chang Su, Tao Zhang

**Affiliations:** 1Department of Biostatistics, School of Public Health, Cheeloo College of Medicine, Shandong University, Jinan 250012, China; 201915766@mail.sdu.edu.cn (C.G.); wenhaoyu@mail.sdu.edu.cn (W.Y.); chunxiali@mail.sdu.edu.cn (C.L.); fanbingbing92@163.com (B.F.); jialilv@mail.sdu.edu.cn (J.L.); weimengke1012@126.com (M.W.); corey0607@163.com (L.H.); 2Institute for Medical Dataology, Shandong University, Jinan 250002, China; 3National Institute of Health Data Science of China, Jinan 250002, China; 4Maternal and Child Health Care of Shandong Province, Cheeloo College of Medicine, Shandong University, Jinan 250012, China; zt2730@hotmail.com; 5National Institute for Nutrition and Health, Chinese Center for Disease Control and Prevention, Beijing 102206, China

**Keywords:** alcohol consumption, body mass index, interaction effect, lipid profiles, China health

## Abstract

**Background:** Both obesity and alcohol consumption are strongly associated with dyslipidemia; however, it remains unclear whether their joint effect on lipid profiles is through mediation, interaction, or a combination of the two. **Methods:** In total, 9849 subjects were selected from the 2009 panel of China Health and Nutrition Survey (CHNS). A four-way decomposition method was used to validate the pathways of drinking and body mass index (BMI) on lipids (total cholesterol, TC; triglyceride, TG; low-density lipoprotein cholesterol, LDL-C; high-density lipoprotein cholesterol, HDL-C; apolipoprotein A, APO-A; and apolipoprotein B, APO-B). **Results:** According to four-way decomposition, the total effects of drinking on lipids were found to be statistically significant, except for LDL-C. The components due to reference interaction were 0.63, 0.48, 0.60, −0.39, −0.30, and 0.20 for TC, TG, LDL-C, HDL-C, APO-A and APO-B, respectively (*p* < 0.05 for all). The effect size of pure indirect effect and mediated interaction were 0.001~0.006 (*p* > 0.05 for all). Further, linear regression models were used to examine the effect of BMI on lipid profiles in drinkers and non-drinkers. The associations of BMI and lipids were higher in all drinkers than in non-drinkers (0.069 versus 0.048 for TC, 0.079 versus 0.059 for TG, 0.057 versus 0.037 for LDL-C, −0.045 versus −0.029 for HDL-C, −0.024 versus −0.011 for APO-A and 0.026 versus 0.019 for APO-B, *p* interaction <0.05 for all). **Conclusions:** The joint effect of alcohol consumption and obesity on lipid profiles is through interaction rather than mediation. Alcohol consumption amplifies the harmful effect of BMI on lipid profiles. Greater attention should be paid to lipid health and cardiovascular risk in obese individuals regarding alcohol consumption. For obese individuals, we do not recommend alcohol consumption.

## 1. Introduction

Dyslipidemia is an important risk factor for the development of atherosclerotic cardiovascular disease [1]. Prevention of dyslipidemia represents the key issue in cardiovascular disease (CVD) prophylaxis. Modifiable factors, such as obesity and drinking, contribute to the accelerated epidemic of dyslipidemia.

Obesity increases the risk of several health conditions including dyslipidemia, hypertension, and type 2 diabetes, fostering a growing global health concern [2]. Due to enormous economic and dietary transformations, the prevalence of obesity has rapidly increased among the Chinese population [3]. The association of obesity with lipid parameters has been well established in many observational studies [4,5,6,7,8]. In obese individuals, adipocytes may release higher levels of free fatty acids (FFA), reducing the clearance of TG-rich lipoproteins in plasma, further facilitating TG release and inhibiting the chylomicrons for lipolysis [8]. An intervention study has additionally found that long-term weight loss can improve lipid profiles [9].

Obesity may be affected by other lifestyle factors, including alcohol consumption and smoking. Alcohol consumption plays an important role in the process of weight gain [10,11,12,13,14,15], as it provides a potential source of energy just after fat [16]. Alcohol consumption is a well-known determinant of serum lipids levels [17,18,19]. Moderate alcohol consumption appears to raise levels of high-density lipoprotein cholesterol (HDL-C) and triglyceride (TG) while lowering levels of low-density lipoprotein cholesterol (LDL-C), whereas heavy alcohol consumption has an unfavorable effect on serum lipids and lipoproteins [20]. It is reasonable to assume that the manner of alcohol consumption could contribute to the development of obesity, and further changes to lipids parameters causing dyslipidemia. Moreover, studies suggest both obesity and alcohol intake affect the process of lipoprotein lipase (LPL) generation, which plays an important role in the pathogenesis of dyslipidemia [19,21]. However, there is currently a paucity of literature that focuses on the interplay between obesity and alcohol consumption and its effects on dyslipidemia. Whether there is an interaction between alcohol intake and obesity on lipids, or whether obesity mediates the association between alcohol consumption and lipid profile, remains unclear.

Traditional mediation analyses and regression models with interaction terms cannot evaluate the mediation effect and interaction simultaneously. In addition, if the exposure and mediator have an interaction on the outcome, the type I error of traditional mediation methods could be increased according to our simulation study. The four-way decomposition provides maximum insight into the level of effect mediated, to which extent this effect is due to interaction, or to both mediation and interaction together, and how much is due to neither. Based on four-way decomposition, this study aims to explore whether the effect of alcohol consumption on lipids is mediated by BMI, whether the impact of BMI on lipids is modified by patterns of alcohol consumption, or both.

## 2. Methods

### 2.1. Study Population

The China Health and Nutrition Survey (CHNS) is an ongoing longitudinal cohort project implemented by national and local governments. It is designed to understand how the social and economic transformation of Chinese society affects the health and nutritional status of the Chinese population. A multi-stage, random cluster data collection process was used to collect data from Beijing, Chongqing, Guangxi, Guizhou, Heilongjiang, Henan, Hubei, Hunan, Jiangsu, Liaoning, Shaanxi, Shandong, Shanghai, Yunnan, and Zhejiang. A total of nine cross-sectional surveys were completed during 1989~2015, covering 4400 households with 33,348 individuals. Details of the study design and sampling strategies are available online (https://www.cpc.unc.edu/projects/china, (accessed on 26 November 2019)) and in published studies [22,23,24].

The biomarker data were only collected in the 2009 CHNS survey; thus, we used the adult subjects of 2009 CHNS survey for our analysis (*n* = 10,076). In total, 227 participants who were diagnosed with stroke or myocardial infarction were excluded. Ultimately, we included 9849 subjects for analysis (4687 males and 5162 females; mean age = 49.87 (15.49) years, with a range = 18.0~98.9 years). In each analysis, adult subjects who had missing values in current lipid-related biomarkers, BMI, age, energy, region, education, and drinking were also excluded.

Study protocols were approved by the Institutional Review Committees of the University of North Carolina at Chapel Hill, NC, USA, and the China National Institute of Nutrition and Food Safety at the Chinese Center for Disease Control and Prevention, Beijing, China. Written informed consent was obtained from each study participant.

### 2.2. Measurements

Standardized protocols were used by trained examiners. Standing height was measured without shoes to the nearest 0.2 cm using a portable SECA stadiometer (SECA, Hamburg, Germany). Weight in light clothing without shoes was measured to the nearest 0.1 kg on a dedicated scale that was routinely calibrated. BMI was calculated as weight in kilograms divided by height in meters squared. 

Serum total cholesterol (TC), triglyceride (TG), high-density lipoprotein cholesterol (HDL-C) and low-density lipoprotein cholesterol (LDL-C) were detected by using CHOD-PAP, Kyowa (Japan). Serum apolipoprotein A (APO-A) and apolipoprotein B (APO-B) were detected using an immunoturbidimetric method, Randox (UK). The calibrators and control serums were provided by the Department of Laboratory Medicine of the China–Japan Friendship Hospital and had the same lot number.

“Living in urban area” indicated whether the individual lived in an urban area or not. Educational attainment was classified as: lower than/graduated from primary school, lower/upper middle school degree, technical or vocational degree, university or college degree, master’s degree or higher. Information on medication history and behavioral lifestyles was obtained in a questionnaire survey. Average daily energy intake was obtained from the dietary intake data collected in the CHNS, which were derived from the Chinese food composition table. For alcohol consumption, individuals were asked three questions in CHNS: “Have you consumed alcohol during the past year (yes, no)?”, “Choose type of alcohol you drink: beer, grape wine (including various colored wines, rice wine) and liquor”, and “For each type of alcohol, how much do you drink each week?” For each type of alcohol, weekly alcohol intake (g) was calculated as the sum of the product of the consumption and alcoholicity. The alcoholicity for beer, grape wine and liquor are 0.034 g/mL, 0.103 g/mL, and 0.36 g/mL, respectively. The total weekly alcohol intake was the sum of weekly alcohol intake for each type of alcohol. The definition of alcohol drinking was defined as 25 g or more per week for men and 15 g or more per week for women.

### 2.3. Statistical Methods

Differences in characteristics across drinking categories were tested for significance. The unpaired t-test or Mann–Whitney U-test was used to compare the differences between continuous variables, and the χ^2^ test was used for categorical variables. Log transformation was applied for TG because of its positive skewness. Four-way decomposition proposed by VanderWeele et al. [25] was used to validate the relationship of drinking, BMI, and lipids. The overall effect of an exposure on an outcome, in the presence of a mediator with which the exposure may interact, can be decomposed into four components, shown as the equation in the following: (i) the effect of the exposure in the absence of the mediator (CDE), (ii) the interactive effect when the mediator is left to what it would be in the absence of exposure (INT_ref_), (iii) a mediated interaction (INT_med_), which is significant only if there is an interaction between mediator and exposure, at the same time the exposure affects the mediator, and (iv) a pure mediated effect (PIE). These four components, respectively, correspond to the portion of the effect that is due to neither mediation nor interaction, to interaction (but not mediation), to both mediation and interaction, and to mediation (but not interaction). The relationship among exposure X, potential mediator M, and outcome Y is shown in Figure 1.
$$TE=CDE+{INT}_{ref}+{INT}_{med}+PIE$$

The modification effect of drinking on the association between BMI and lipid-related biomarkers was confirmed by linear regression models with interactions. Further, we detect the effect of BMI on lipid parameters at different drinking status using linear regression models. Age, energy intake, region, and education were adjusted as covariates in four-way decomposition and regression models. Statistical analyses were implemented with R for windows (v. 3.6.3, http://www.r-project.org/ (accessed on 25 November 2021)) and SAS version 9.4 (SAS Institute, Cary, NC, USA).

## 3. Results

Table 1 summarizes descriptive characteristics of study variables by drinking status. Continuous variables were compared between drinkers and non-drinkers, adjusting for age (except age itself). The mean levels of alcohol consumption, TC, TG, APO-A, APO-B, and energy intake were significantly higher in drinkers, while non-drinkers had higher LDL-C. In total, 77.7% of non-drinkers have never drunk, while over a half of drinkers drink every day. In addition, the proportion of males was higher among drinkers than among non-drinkers; the distribution of educational level had significant group differences.

Table 2 shows the effect of drinking on lipids due to mediation and interaction with BMI. We found that the total effect and the reference interactions were all statistically significant (except for LDL-C). The total effects of drinking on lipids were 0.214 for TC, (*p* < 0.001), 0.091 for TG (*p* < 0.001), −0.011 for LDL-C (*p* = 0.765), 0.117 for HDL-C (*p* < 0.001), 0.108 for APO-A (*p* < 0.001), and 0.034 for APO-B (*p* = 0.001). The effect size of reference interactions (INT_ref_) among the four components in the six biomarkers was relatively large: 0.625 (*p* = 0.012), 0.483 (*p* = 0.002), 0.595 (*p* = 0.014), −0.387 (*p* = 0.002), −0.295 (*p* = 0.001), and 0.198 (*p* = 0.002) separately for TC, TG, LDL-C., HDL-C, APO-A, and APO-B; the mediated interaction and the pure mediation were both insignificant and the effect sizes were small.

Figure 2 shows the effect of BMI on lipid-related biomarkers in drinkers and non-drinkers. The associations of BMI and lipids were all higher in drinkers than in non-drinkers (0.069 versus 0.048 for TC, 0.079 versus 0.059 for TG, 0.057 versus 0.037 for LDL-C, −0.045 versus −0.029 for HDL-C, −0.024 versus −0.011 for APO-A, and 0.026 versus 0.019 for APO-B; *p* values for these differences were all <0.05).

Table 3 shows the standardized regression coefficients of BMI for lipids in different drinking groups. The regression coefficients of BMI on TC, Ln-TG, LDL-C, and APO-B were positive, and were higher in drinkers than in non-drinkers, both for males and females (0.24 versus 0.20 for TC in males, 0.24 versus 0.14 for TC in females; 0.45 versus 0.38 for Ln-TG in males; 0.33 versus 0.27 for Ln-TG in females; 0.19 versus 0.13 for LDL-C in males, 0.34 versus 0.12 for LDL-C in females; 0.33 versus 0.27 for APO-B in males, 0.42 versus 0.21 for APO-B in females). The regression coefficients of BMI on HDL-C and APO-A were negative, and were higher in drinkers than non-drinkers, both for males and females (−0.32 versus −0.24 for HDL-C in males, −0.30 versus −0.18 for HDL-C in females; −0.24 versus −0.13 for APO-A in males, −0.13 versus −0.08 for APO-A in females).

## 4. Discussion

In the current study, our findings confirmed obesity and drinking may influence the development of dyslipidemia. Moreover, we additionally provided consolidated evidence that there was only a reference interaction (INT_ref_) between BMI and alcohol consumption on the changing of lipid profiles, and alcohol consumption amplified the association between BMI and lipids.

It is almost certain that obesity has an unfavorable effect on lipids [4,5,6,7,8]. The adipocytes of obese individuals become hypertrophic and participate in metabolic regulation through endocrines [8], and dyslipidemia is a common obesity-related metabolic abnormality. Studies have found that the main signs of dyslipidemia in obese patients are increased triglyceride levels and decreased high-density lipoprotein levels [19,26,27]. We also noted a positive correlation between BMI and TG and a negative correlation between BMI and HDL-C. In addition, our findings revealed BMI was positively associated with TC, LDL-C, and APO-B, while being inversely associated with APO-A. The effect of BMI on lipids is stronger in drinkers than in non-drinkers, both in males and females.

The association between drinking and plasma lipid levels has been studied for decades and remains controversial [28]. It has been shown that moderate alcohol consumption may increase levels of HDL-C, APO-A, and decrease levels of LDL-C [18]. This is consistent with our results. However, a Japanese study found that drinking is associated with high levels of TC and TG [29]. We could also draw the above conclusions through Table 1 and Table 2. This relationship is explained by alcohol consumption’s damaging effects on cholesterol homeostasis, inhibiting the oxidation of free fatty acids (FFA), influencing the selective enrichment of polyunsaturated fatty acids, or by inhibiting the enzymatic activity of the cholesteryl ester transfer protein [17,30,31,32,33]. 

There are several putative mechanisms linking adiposity and alcohol assumption to lipids. Moreover, there is growing evidence that insulin resistance (IR) plays a key role in obesity-related metabolic disorders. In the state of IR, adipocytes release more FFA, and the clearance of TG-rich lipoproteins in plasma is further reduced, causing TG release. Further, the increased FFA in the plasma of obese patients flows to the liver, forms a large amount of very low-density lipoprotein, and competes with chylomicrons for lipolysis [34]. In addition, obesity may cause hypertriglyceridemia by decreasing the expression of lipoprotein lipase (LPL) mRNA and reducing the lipolytic capacity of triglyceride-rich lipoproteins [19,27]. However, an experiment in mice demonstrated that alcohol consumption can upregulate the expression of LPL [21]. Therefore, obesity and drinking may jointly affect the process of LPL generation, which is subsequently involved in the pathogenesis of obesity-related hypertriglyceridemia. Additionally, obesity-related fat factors such as leptin are reported to be related to the occurrence and development of dyslipidemia. Circulating leptin levels can decrease the intracellular lipid content in several tissues via increases in lipid oxidation [34,35,36]. A mendelian randomization study found that leptin was negatively associated with lipid levels and this association can be modified by alcohol consumption [37]. Based on the four-way decomposition, we drew similar conclusions. There is only an interaction between BMI and drinking habit on lipids. Alcohol consumption amplifies the effect of BMI on lipid profile.

### Strengths and Weakness

We used the CHNS which is a representative survey for Chinese population, so the findings are convincing; in addition, the present analysis confirmed the interaction effect of alcohol consumption on BMI–lipid relationships by the four-way decomposition method, which may provide a novel insight for the prevention and control of dyslipidemia and CVD. Our research also highlights some limitations. Much of the information in this study, such as drinking status, and disease history information, was given by a self-reported questionnaire, which may induce recall bias and misclassification. Additionally, due to the limitation of data collocation, we could not explore the dose–response relationship between alcohol consumption. The information of medication in our questionnaire was limited; thus, the analysis could not completely exclude the influence of medication. Furthermore, as this was a cross-sectional study and there are some unknown confounders that cannot be adjusted for, it is difficult to determine a clear causal effect. The association of alcohol drinking and lipids might represent a false positive finding.

## 5. Conclusions

In summary, these findings suggest obesity has a harmful effect on lipids, and alcohol intake may improve serum HDL-C and APO-A levels but decrease serum TC, TG, and APO-B levels. The joint effect of alcohol consumption and obesity on lipid profiles is through interaction rather than mediation. Alcohol consumption amplifies the effect of BMI on blood lipid concentrations. The association of BMI and lipids profiles is stronger in drinkers than in non-drinkers. Therefore, greater attention should be drawn to lipid health and cardiovascular risk in obese individuals who partake in alcohol consumption. For obese individuals, we do not recommend heavy alcohol consumption. 

## Figures and Tables

**Figure 1 ijerph-18-13211-f001:**
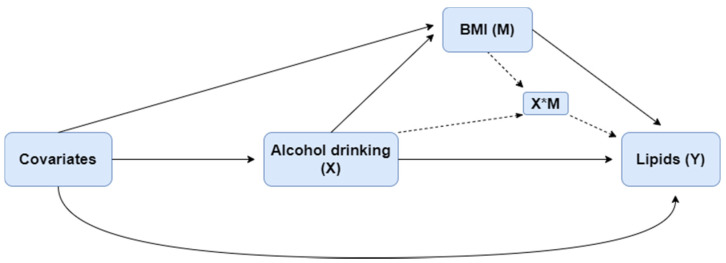
The relationship among alcohol consumption (X), BMI (M), and lipids (Y).

**Figure 2 ijerph-18-13211-f002:**
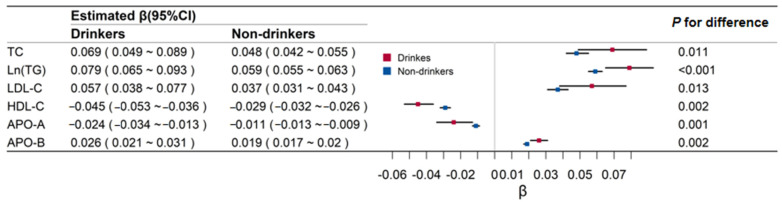
Regression coefficients and 95% confidence intervals (CI) of body mass index for lipids by alcohol drinking groups. TC means total cholesterol; Ln (TG) means Log transformed triglyceride; LDL-C means low-density lipoprotein cholesterol; HDL-C means high-density lipoprotein cholesterol; APO-A means apolipoprotein A; APO-B means apolipoprotein B.

**Table 1 ijerph-18-13211-t001:** Descriptive data of study variables by alcohol drinking.

Variable	Non-Drinker (*n* = 8430)	Drinker (*n* = 1016)	Total (*n* = 9446)	*p* Value
Age, years	50.0 (15.8)	50.5 (12.5)	50.1 (15.5)	0.401
Males, *n* (%)	3466 (41.1)	913 (89.9)	4379 (46.4)	<0.001
Alcohol consumption, g/week	9.4 (6.1)	55.3 (37.6)	25.5 (31.6)	<0.001
Drinking frequency, *n* (%)				<0.001
Almost every day	278 (3.3)	575 (56.9)	853 (9.0)	
Three or four times a week	210 (2.5)	195 (19.3)	405 (4.3)	
Once or twice a week	560 (6.7)	159 (15.7)	719 (7.6)	
Once or twice a month	543 (6.5)	60 (5.9)	603 (6.4)	
No more than once a month	288 (3.4)	21 (2.1)	309 (3.3)	
Never	6545 (77.7)	0 (0.00)	6545 (77.7)	
BMI, kg/m^2^	23.3 (3.5)	23.5 (3.3)	23.3 (3.5)	0.110
TC, mmol/L	4.9 (1.0)	5.0 (1.0)	4.9 (1.0)	<0.001
TG, mmol/L	1.6 (1.4)	2.0 (1.9)	1.7 (1.5)	<0.001
LDL-C, mmol/L	3.0 (1.0)	2.9 (1.0)	3.0 (1.0)	<0.001
HDL-C, mmol/L	1.4 (0.5)	1.5 (0.53)	1.5 (0.5)	0.093
APO-A, g/L	1.1 (0.4)	1.2 (0.5)	1.2 (0.4)	<0.001
APO-B, g/L	0.9 (0.27)	0.94 (0.28)	0.91 (0.27)	0.002
Energy, kcal	2086.9 (661.2)	2453.5 (719.3)	2126.4 (677.3)	<0.001
Education, *n* (%)				<0.001
Lower than primary school	2013 (23.9)	155 (15.3)	2168 (23.0)	
Graduated from primary school	1623 (19.3)	181 (17.8)	1804 (19.1)	
Lower middle school degree	2761 (32.8)	404 (39.8)	3165 (33.6)	
Upper middle school degree	984 (11.7)	136 (13.4)	1120 (11.9)	
Technical or vocational degree	607 (7.2)	74 (7.3)	681 (7.2)	
University or college degree	424 (5.0)	65 (6.4)	489 (5.2)	
Master’s degree or higher	6 (0.1)	0 (0.0)	6 (0.1)	
Region, *n* (%)				0.586
Urban	2848 (33.8)	334 (32.9)	3182 (33.7)	
Rural	5582 (66.2)	682 (67.1)	6264 (66.3)	

Continuous variables are presented as means (SD); *p* values were adjusted for age.

**Table 2 ijerph-18-13211-t002:** The effect of drinking on lipids due to mediation and interaction with BMI.

	Total Effect		CDE		INT_ref_		INT_med_		PIE	
	Est (SE)	*p* Value	Est (SE)	*p* Value	Est (SE)	*p* Value	Est (SE)	*p* Value	Est (SE)	*p* Value
TC	0.214 (0.038)	<0.001	−0.421 (0.251)	0.094	0.625 (0.247)	0.012	0.004 (0.004)	0.350	0.006 (0.006)	0.316
Ln (TG)	0.091 (0.025)	<0.001	−0.403 (0.155)	0.009	0.483 (0.152)	0.002	0.003 (0.003)	0.346	0.008 (0.008)	0.324
LDL-C	−0.011 (0.037)	0.765	−0.615 (0.247)	0.013	0.595 (0.243)	0.014	0.003 (0.004)	0.348	0.005 (0.005)	0.312
HDL-C	0.117 (0.019)	<0.001	0.509 (0.124)	<0.001	−0.387 (0.122)	0.002	−0.002 (0.002)	0.338	−0.004 (0.004)	0.316
APO-A	0.108 (0.014)	<0.001	0.405 (0.092)	<0.001	−0.295 (0.091)	0.001	−0.002 (0.002)	0.337	−0.001 (0.001)	0.318
APO-B	0.034 (0.010)	0.001	−0.168 (0.066)	0.011	0.198 (0.065)	0.002	0.001 (0.001)	0.340	0.002 (0.002)	0.315

CDE = controlled direct effect of drinking on lipids; INT_ref_ = the reference interaction of drinking and BMI; INT_med_ = the mediated interaction of drinking and BMI; PIE = the pure indirect effect of drinking, BMI and lipids; TC = total cholesterol; Ln(TG) = Log transformed triglyceride; LDL-C = low-density lipoprotein cholesterol; HDL-C = high-density lipoprotein cholesterol; APO-A = apolipoprotein A; APO-B = apolipoprotein B; *p* values were adjusted for age, energy, education, region, and sex.

**Table 3 ijerph-18-13211-t003:** The standardized regression coefficients of body mass index for lipids by different drinking groups in males and females.

	Males			Females		
	Drinkers	Non-Drinkers	*p* for Difference	Drinkers	Non-Drinkers	*p* for Difference
TC	0.24	0.20	0.126	0.24	0.14	0.439
Ln-TG	0.45	0.38	0.045	0.33	0.27	0.861
HDL-C	−0.32	−0.24	0.073	−0.30	−0.18	0.159
LDL-C	0.19	0.13	0.102	0.34	0.12	0.037
APO-A	−0.24	−0.13	0.034	−0.13	−0.08	0.328
APO-B	0.33	0.27	0.071	0.42	0.21	0.072

The *p* values of the estimated standardized regression coefficients were all <0.05; age, energy, education, and region were included in the regression models; TC means total cholesterol; Ln (TG) means Log transformed triglyceride; LDL-C means low-density lipoprotein cholesterol; HDL-C means high-density lipoprotein cholesterol; APO-A means apolipoprotein A; APO-B means apolipoprotein B.

## Data Availability

The dataset supporting the conclusions of this article is available in a public, open access repository, https://www.cpc.unc.edu/projects/china/data (accessed on 26 November 2019).

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
