# Peer review of "Four-Way Decomposition of Effect of Alcohol Consumption and Body Mass Index on Lipid Profile"

_ijerph, 2021, doi:10.3390/ijerph182413211_

Round 1
Reviewer 1 Report
Authors presents a secondary analysis about the association between alcohol intake, lipid metabolism, and obesity. Variance analysis is applied to investigate the direct, obesity mediated, obesity interacted, and indirect effect of alcohol on lipid status. The demonstration uses a database built up in China. The topic is important.
Introduction:
The trivial importance of obesity and lipid status, and the association between these factors is not completed with the short summary on alcohol intake’s importance. The alcohol drinking as a life style factor which is covaried with other (nutritional and physical activity related) life style element associated with weight gain is not summarized. The alcohol metabolic effects and the alcohol life style effects are not separated from each other, and the preexisting knowledge on these independent mechanisms are not presented. The introduction does suggest that the alcohol effect is a biochemical issue.
The interaction between alcohol intake and BMI on lipid status is supported by one reference from 2012. Much more thorough literature search is required to demonstrate the preexisting and the lacking knowledge.
Terms such “additional alcohol consumption” (line 41), “behavioral alcohol drinking” (line 45) are unusual, should be replaced with the standard terms.
Typo (?) “study in Japanese 48 man” (line 48-49)
Typo “Therefore, Alcohol” (line 50)
Objectives:
Different causal mechanisms are mentioned in this section, which are defined later in the text (in the Methods section). Because the four-way decomposition is not among the frequently used statistical approaches (readers can be not familiar with that), the causal mechanisms should be explained before formulating questions using those. In the introduction, the lack of knowledge and the usefulness of the four-way decomposition analysis to answer the research question could be presented.
Methods:
The survey which produced the database from 2009 for analysis is presented in a superficial manner. Neither the population represented is introduced, nor the sampling are described. There is no reference for this section at all. (line 73)
Exclusion criteria contained parameters which are utilized in the analysis. Why? Number of excluded cases is not reported. It would be much more important than to describe the laboratory methods. (line 79-90) (e.g.: Is it really important to describe that „three 4ml tubes” were used?)
These sentences should be reformulated:
“Adult subjects who were diagnosed as stroke or myocardial infarction were excluded (n=227).” (line 68-69)
“Geographical regions were divided into rural and urban.” (line 91)
The term “medication history” is about drug consumption or about personal history? (line 94)
Typo (line 95) “intake(kcal)”
Exposure to alcohol was assessed as “Alcohol consumption was defined as subjects ever drunk beer or any other alcoholic beverage last year.” (line 97) It is the main parameter investigated. Its quantification should be presented in detail. Does it mean that ANY (even very small) amount of alcohol intake a year was enough to be classified as alcohol drinker? The huge literature on alcohol’s effects demonstrate that the relevant dose-response relationships are not monotonous. The small doses has no harmful impact on the cardio-metabolic status.
Grammar (line 100): “Analyses of covariance were performed”
Regarding Statistical methods section:
A casual path diagram should be prepared (as it was presented in the methodological paper cited by authors).
It is not explained how were the confounding effects of education, rural/urban residential place, energy intake, sex and age controlled for in the variance analysis. Was it at all?
The distributions of the parameters used in the variance analysis are not presented. In spite the fact that the normal distribution is pre-requirement.
Regression model (or medels?) are not described properly. (line 115-118)
Result:
There are two paragraphs on the presentation of findings. Taken into consideration the complexity of the statistical approach, and the number of statistical outputs, it seems to be inadequate.
Because “Log transformation was applied for TG because of its positive skewness.” (line 101), it should be clarified whether the record is about TC or lnTC.
Discussion:
How is it possible that the total effect of alcohol intake on LDL is not significant while there are significant impacts through CDE, INTref, and PIE?
The regression coefficients among drinkers and non-drinkers have overlapping 95% confidence interval and significant test results for their difference. What does it mean?
No discussion about the role of weak exposure assessment and the lack of confounding factors’ control.
Conclusions:
If the results are valid (according to the missing evaluation of the validity issues) then the conclusions are establish
Reviewer 2 Report
The manuscript of Chaonan Gao et al, is dedicated to the effect of alcohol consumption and BMI on lipid profile based on the analysis of China Health and Nutrition Survey data . This is well written and interesting to readers manuscript, but the following thoughts prevent my full appreciation of it:
- In Abstract lines 29-30 is mentioned “The joint effect of alcohol consumption and obesity on lipid profiles is interaction other than mediation”. Authors must specify what kind of interaction they have observed but not just “other that mediation” The same conclusion should be done in the Discussion.
- In Introduction, please provide the information about unsolved problems that you are going to solve with your study.
- In lanes 97-98 is mentioned “Alcohol consumption was defined as subjects ever drunk beer or any other 97 alcoholic beverage last year”. This is not enough for analysis. Please provide information about amount, frequency and type of alcohol consumed.
- Because of possible sex-dependent response to the alcohol it is necessary to split the patients based on sex and analyze the results separately.
- It is necessary to reorganize "Discussion" with clear summary of obtained results and highlighting what new knowledge you are bringing to scientific community.
Round 2
Reviewer 1 Report
All of my comments have been reflected properly by authors apart from the question about the quantification of the exposure to alcohol.
In the original manuscript it was written: "Alcohol consumption was defined as subjects ever drunk beer or any other alcoholic beverage last year."
In the revised manuscript, this sentence has been replaced: "The definition of alcohol drinking was defined as 25g or more per week for men and 15g or more per week for women. "
Although, authors state that these two definitions are equivalent, these definitions are obviously different. Since alcohol intake is the most important parameter of the paper (all the results and the conclusive sentences are determined by the reliability of this parameter), the method applied in the assessment has to be described clearly. The inadequate reflection to this crucial point is in sharp opposition with the adequacy of all other responses. Questions used in alcohol intake assessment with references should be added to the final version of the description for alcohol intake assessment.
Reviewer 2 Report
Thank you very much for your responses to my comments.
The manuscript improved significantly and may be accepted in present form.
Author Response
Thank you for your approval. We will continue our efforts!